# A Variety of Mouse PYHIN Proteins Restrict Murine and Human Retroviruses

**DOI:** 10.3390/v16040493

**Published:** 2024-03-23

**Authors:** Sümeyye Erdemci-Evin, Matteo Bosso, Veronika Krchlikova, Wibke Bayer, Kerstin Regensburger, Martha Mayer, Ulf Dittmer, Daniel Sauter, Dorota Kmiec, Frank Kirchhoff

**Affiliations:** 1Institute of Molecular Virology, Ulm University Medical Center, 89081 Ulm, Germany; suemeyye.erdemci-evin@uni-ulm.de (S.E.-E.); matteo.bosso@med.uni-tuebingen.de (M.B.); kerstin.regensburger@uni-ulm.de (K.R.); martha.mayer@uni-ulm.de (M.M.); dorota.kmiec@uni-ulm.de (D.K.); 2Institute for Medical Virology and Epidemiology of Viral Diseases, University Hospital Tübingen, 72076 Tübingen, Germany; veronika.krchlikova@med.uni-tuebingen.de (V.K.);; 3Institute for Virology, University Hospital Essen, University of Duisburg-Essen, 45147 Essen, Germany; wibke.bayer@uk-essen.de (W.B.); ulf.dittmer@uk-essen.de (U.D.)

**Keywords:** murine PYHIN proteins, HIV-1, mouse leukemia virus, retroviral transcription

## Abstract

PYHIN proteins are only found in mammals and play key roles in the defense against bacterial and viral pathogens. The corresponding gene locus shows variable deletion and expansion ranging from 0 genes in bats, over 1 in cows, and 4 in humans to a maximum of 13 in mice. While initially thought to act as cytosolic immune sensors that recognize foreign DNA, increasing evidence suggests that PYHIN proteins also inhibit viral pathogens by more direct mechanisms. Here, we examined the ability of all 13 murine PYHIN proteins to inhibit HIV-1 and murine leukemia virus (MLV). We show that overexpression of p203, p204, p205, p208, p209, p210, p211, and p212 strongly inhibits production of infectious HIV-1; p202, p207, and p213 had no significant effects, while p206 and p214 showed intermediate phenotypes. The inhibitory effects on infectious HIV-1 production correlated significantly with the suppression of reporter gene expression by a proviral Moloney MLV-eGFP construct and HIV-1 and Friend MLV LTR luciferase reporter constructs. Altogether, our data show that the antiretroviral activity of PYHIN proteins is conserved between men and mice and further support the key role of nuclear PYHIN proteins in innate antiviral immunity.

## 1. Introduction

Pyrin and HIN domain-containing (PYHIN) proteins play complex roles in innate immunity [1,2,3]. As the best-characterized member of this protein family is AIM2 (Absent In Melanoma 2), they are also known as AIM2-like receptors (ALRs). Human AIM2 acts as a cytosolic immune sensor that triggers inflammasome activation upon detection of pathogen-derived double-stranded DNAs [4,5,6]. Humans encode three additional PYHIN proteins: γ-IFN-inducible protein 16 (IFI16), IFN-inducible protein X (IFIX), also known as Pyrin and HIN domain-containing protein 1 (PYHIN1), and Myeloid Nuclear Differentiation Antigen (MNDA) [7]. All four human PYHIN proteins contain an N-terminal Pyrin domain (PYD) and one or two C-terminal hematopoietic interferon-inducible nuclear proteins with a 200-amino-acid repeat domain (HIN200). PYDs belong to the superfamily of death domains (DDs) and promote interactions with other PYD-containing proteins to regulate inflammation, apoptosis, and cell cycling. The HIN domain is characteristic of PYHIN proteins. It mediates DNA binding by electrostatic interactions between positively charged amino acid residues in the HIN domain and the negatively charged phosphate groups in the DNA backbone [8,9,10]. Upon DNA binding, AIM2 mediates the assembly of inflammasomes, initiating activation of caspase-1 and subsequently the production of proinflammatory cytokines, including interleukin-1β and -18 [4,11]. Thus, AIM2 is thought to play a key role in the activation of antiviral effectors and recruitment of immune cells at sites of pathogen exposure.

Initially, it was assumed that human PYHIN1, IFI16, and MNDA may also act as cytosolic sensors of foreign DNA that trigger immune responses against viral or bacterial pathogens. Especially for IFI16, numerous studies reported roles in the sensing of pathogen-derived nucleic acids in both the cytosol and the nucleus [12,13,14,15,16,17]. However, its role as an innate DNA sensor is under debate. In some settings, IFI16 promotes immune activation upon cGAS-STING-mediated DNA sensing rather than directly acting as a DNA sensor [16]. In contrast to AIM2, PYHIN1, IFI16, and MNDA contain nuclear localization signals (NLSs) in a linker region connecting their Pyrin and HIN domains [7]. Thus, they are usually predominantly found in the nucleus and were reported to inhibit the transcription of different viral pathogens by various sensing-independent mechanisms, including epigenetic modifications, occupation of viral DNAs, and sequestration of the transcription factor Sp1 [1].

Mice harbor a striking number of 13 PYHIN-coding genes that arose from sequence duplications and inversions on chromosome 1 [7,18]. In addition, a short Pyrin-only splice variant of mouse AIM2/p210, named AIM2b/p210b, has been reported [7]. Mice are commonly used as model organisms for human immune responses, and early studies reported the roles of murine PYHIN proteins in innate dsDNA sensing and inflammasome activation [19,20,21]. Thus, it came as a surprise that the deletion of all 13 PYHIN genes in mice had no significant effect on the IFN response to transfected DNA ligands, DNA virus infection, and lentivirus infection [22]. This finding suggested that immune sensing may not be the major function of PYHIN proteins. Indeed, the lack of the PYHIN locus was associated with higher levels of Friend virus viremia during acute infection, and it has been shown that the mouse PYHIN protein p204 restricts retroviral replication [23]. However, the potential antiretroviral functions of the remaining 12 murine PYHIN proteins remain to be determined. Here, we show that most mouse PYHIN proteins share the antiretroviral activity of human nuclear PYHIN proteins. Thus, our results demonstrate the functional redundancy of murine PYHIN proteins in restricting murine and human retroviruses.

## 2. Materials and Methods

*Cell Lines.* HEK293T and TZM-bl cells were maintained in Dulbecco’s Modified Eagle Medium (DMEM) supplemented with 10% FCS, glutamine (2 mM), streptomycin (100 μg/mL), and penicillin (100 U/mL). HEK293T cells were provided and authenticated by the ATCC. They were originally isolated from a female fetus and engineered to express the SV40 large T antigen [24]. TZM-bl cells were provided by the NIH AIDS Reagent Program, Division of AIDS, NIAID, NIH from Dr. John C. Kappes, Dr. Xiaoyun Wu, and Tranzyme Inc. TZM-bl were derived from HeLa cells, which were isolated from a 30-year-old female. TZM-bl cells were engineered to express CD4, CCR5, and CXCR4 and carry the β-galactosidase genes under the regulation of the HIV-1 LTR promoter [25]. Both cell lines were maintained in a humidified atmosphere with 5% CO_2_ at 37 °C.

*Analysis of mammalian PYHIN gene loci.* The sequences, names, and orientations of selected genes from mammalian PYHIN gene loci were obtained from genome assemblies published in the NCBI database [human–GRCh38.p14 (GCF_000001405.40); chimpanzee–NHGRI_mPanTro3-v1.1-hic.freeze_pri (GCF_028858775.1); orangutan–NHGRI_mPon-Abe1-v1.1-hic.freeze_pri (GCF_028885655.1); gibbon–Asia_NLE_v1 (GCF_006542625.1); macaque–Mmul_10 (GCF_003339765.1); marmoset–mCalJa1.2.pat.X (GCF_011100555.1); lemur–mLemCat1.pri (GCF_020740605.2); bat–CSHL_Meso_final (GCF_021234165.1); mouse–GRCm39 (GCF_000001635.27); rat–mRatBN7.2 (GCF_015227675.2); cat–F.catus_Fca126_mat1.0 (GCF_018350175.1); elephant–Loxafr3.0 (GCF_000001905.1); armadillo–mDasNov1.hap2 (GCF_030445035.1); opossum–mMonDom1.pri (GCF_027887165.1); tasmanian devil–mSarHar1.11 (GCF_902635505.1)]. The phylogenetic tree was generated using TimeTree [26] using a list of selected mammalian species.

*PYHIN expression plasmids and proviral constructs.* pCG-based vectors co-expressing most human and mouse PYHIN proteins and the blue fluorescent protein (BFP) via an IRES element were generated as previously described [23,27]. Expression constructs for IFIX/PYHIN1 and MNDA were a gift from Professor M. R. Jakobsen. For easier detection of the expressed proteins, a C-terminal HA-tag was introduced via the reverse PCR primer. All constructs were sequenced to validate their correctness. The pCR-XL-TOPO_HIV-1 M subtype B CH058 proviral construct and the *env*-deficient Moloney MLV eGFP reporter virus have been previously reported [23,28].

*Western Blots.* To assess protein expression levels, transfected HEK293T cells were washed with 1 mL of phosphate-buffered saline (PBS) and subsequently lysed in Co-IP lysis buffer containing protease inhibitor cocktail. The lysis buffer composition involved 150 mM NaCl, 50 mM HEPES, 5 mM EDTA, 0.1% NP40, 500 μM Na_3_VO_4_, and 500 μM NaF at pH 7.5. The cell lysates were then subjected to centrifugation at 14,000 rpm for 30 min at 4 °C. The resulting supernatants were mixed with 4× Protein Sample Loading Buffer (catalog no. 928-40004, LI-COR) supplemented with 10% β-mercaptoethanol (Sigma Aldrich, Taufkirchen, Germany). The protein samples were subsequently heated at 95 °C for 5 min and resolved on NuPAGE 4–12% Bis-Tris Gels (Invitrogen, Carlsbad, CA, USA) under constant voltage conditions (120 V) for 80 min. Following gel electrophoresis, the proteins were transferred onto Immobilon-FL PVDF membranes (Thermo Fisher, MA, USA). The membranes were then stained with primary antibodies against specific target proteins, such as HA-tag (Abcam #ab18181) and GAPDH (Santa Cruz #sc-365062), followed by incubation with secondary antibodies (LI-COR IRDye, 1:20,000 dilution). Protein band intensities were measured using Image Studio Software following the detection with secondary antibodies and background substracted.

*Transfections and Virus Stocks.* For transient transfection, HEK293T cells were seeded at a density of 3 × 10^5^ cells per well in 12-well plates containing 1 mL of complete medium. After 24 h, the cells were transfected with proviral constructs in combination with an expression construct for a specific PYHIN protein or empty control vector using the calcium phosphate precipitation method. The transfection medium was replaced within 6 to 16 h post-transfection. At 40 h post-transfection, both the supernatants and cells were harvested, with subsequent centrifugation at 3000× *g* for 3 min. The viral supernatants were stored at 4 °C for a maximum of one week until usage in TZM-bl infection assays, while the cells were stored at −20 °C.

*Viral Infectivity Assay.* To assess the infectious yield of HIV-1, TZM-bl reporter cells were seeded in triplicates at a density of 8000 cells per well in 96-well plates. Subsequently, the cells were infected with the virus stocks in triplicates. After a 72 h incubation period, the supernatants were removed, and the cells were lysed using the GalScreen Kit (Applied Biosystems, Carlsbad, CA, USA) according to the manufacturer’s instructions. The β-galactosidase reporter gene expression was determined by employing an Orion microplate luminometer (Berthold, Pforzheim, Germany).

*Quantitative reverse transcription polymerase chain reaction (qRT-PCR).* The quantification of viral *env* transcripts was performed in HEK293T cells that were cotransfected with the proviral HIV-1 CH058 construct (1.25 μg) and expression vectors for human and murine PYHINs. At 40 h post-transfection, the cells were washed with phosphate-buffered saline (PBS), and total RNA was isolated using the Zymo Research Quick-RNA Microprep Kit (Cat# R1050) following the manufacturer’s instructions. Subsequently, the isolated RNA was utilized for qRT-PCR analysis employing the TaqMan™ Fast Universal PCR Master Mix (ThermoFisher) and viral primer/probes (Biomers, Ulm, Germany/TIB Molbiol, Berlin, Germany) in multiplex reactions, with GAPDH (Thermo Fisher Weltham, MA, USA) serving as the internal control. The viral primers and probes were designed based on previously described sequences [29].

*Luciferase Assay.* For the luciferase assay, poly-L-lysine-coated 96-well plates were seeded with 22,000 HEK293T cells per well. The cells were transfected in triplicates using the calcium phosphate method. Specifically, the cells were cotransfected with firefly luciferase reporter constructs (5 ng) and expression constructs for PYHINs. Constructs expressing the firefly luciferase under the control of the WT or Sp1 mutant HIV-1 LTRs and the CMV IE-dependent Gaussia luciferase vector used for normalization have been previously described [23]. To generate the Friend MLV LTR luciferase reporter construct, the viral 3′LTR was PCR amplified using primers containing *MluI* and *XhoI* restriction sites for cloning into the pGL3 firefly luciferase reporter vector. Forty hours post-transfection, the cells were lysed, and firefly luciferase activity was measured using the Luciferase Assay Kit (Promega, Madison, Wisconsin, United States) according to the manufacturer’s instructions. Gaussia luciferase was measured in supernatants with priming settings to inject 50 μL of 20 μM coelenterazine in PBS. Basal activity of the wildtype HIV-1 LTR was typically ~5 × 10^6^ RLU/s, and experiments were performed in the absence of Tat expression construct. The luminescence signal was detected using an Orion microplate luminometer (Berthold).

*Confocal microscopy*. Glass coverslips 13 mm in diameter were placed into 24-well plates and coated with poly-L-lysine for 1 h. A total of 50,000 HEK293T cells were seeded per well and transfected on the following day with expression constructs for murine PYHIN proteins containing a C-terminal HA-tag. At 40 h post-transfection, cells were washed with ice-cold PBS and fixed in 4% PFA for 20 min at room temperature (RT), then permeabilized and blocked in PBS 0.5% Triton X-100 5% FCS for 30 min. PYHIN proteins were detected with an HA-tag antibody (Abcam, Cambridge, UK) and Alexa Fluor 647 Goat anti-Mouse IgG (H + L) Secondary Antibody (Sino Biological, Beijing, China). Nuclei were stained with DAPI. The fluorescence signal was captured using an LSM710 confocal microscope (Carl Zeiss, Jena, Germany) and analyzed using Fiji image processing software (ImageJ 2.9).

*Enzyme-linked immunosorbent assay (ELISA).* Virus-containing supernatants (180 μL) were inactivated with 20 μL 1 % Triton and incubated for one hour at 37 °C. High binding plates from Sarstedt (Nümbrecht, Germany) were first coated with 100 μL of anti-p24 monoclonal antibody from ExBio (Vestec, Czech Republic) in DPBS and incubated overnight at RT. After each incubation step, plates were washed three times and subsequently treated with blocking buffer for two hours at 37 °C. For detection, 100 μL each of a serially diluted p24 protein standard from Abcam and the samples were incubated overnight at RT in a humid environment. The ELISA plates were further incubated with 100 μL rabbit anti-p24 serum from Eurogentec (Seraing, Belgium) diluted in an antibody buffer, followed by incubation with 100 μL of a goat polyclonal anti-rabbit IgG HRP-conjugated secondary antibody from Dianova (Hamburg, Germany). Subsequently, 100 μL of SureBlue TMB 1-Component Microwell Peroxidase Substrate from SeraCare Life Sciences was added, and the plates were incubated for 20–25 min at RT on a Compact Digital MicroPlate Shaker by Thermo Scientific (Thermo Fisher, Weltham, MA, USA). The reaction was stopped by adding 50 μL of 0.5 M H_2_SO_4_. The absorbance was measured at 450 nm with a baseline correction at 650 nm using a Vmax kinetic ELISA microplate reader from Molecular Devices (GMI-inc, Ramsey, MN, USA). Finally, p24 concentrations were calculated using Graphpad PRISM 10 (GraphPad Software, California, USA) by interpolating the standard curve with the Sigmoidal, 4PL, X is log (concentration) option.

*Multiple sequence alignments and Nuclear Localization Signal analysis.* Sequences were aligned using Multalin sequence alignment software and Clustal Omega. Nuclear localization signals were predicted using NLStradamus software, which utilizes a Hidden Markov Model (HMM)-based prediction method [30]. Predictions were made using the 4-state HMM static prediction option with a cutoff of 0.6.

*Statistical Analysis.* Statistical analyses were performed using GraphPad PRISM 10 (GraphPad Software, California, USA). *p*-values were determined using a two-tailed Student’s *t*-test. Correlations were calculated with the linear regression module. Unless otherwise stated, data are shown as the mean of at least three independent experiments ± SEM. Significant differences are indicated as follows: *, *p* < 0.05; **, *p* < 0.01; and ***, *p* < 0.001. Statistical parameters are specified in the figure legends.

## 3. Results

### 3.1. Multiplication of PYHIN Genes in Mice

PYHIN genes evolved in mammalian species and have undergone extensive gene expansion and divergence [7,18]. It is thought that initially, an HIN domain emerged in the common ancestor of marsupials and placental mammals. Later, three distinct forms (HIN-A, -B, and -C) evolved in placental mammals [7]. The number of PYHIN genes varies enormously between different mammalian species. Marsupials, such as opossums and Tasmanian devils, harbor only a single PYHIN coding sequence (Figure 1). Cats encode 2 PYHIN proteins, humans 4 (AIM2, IFI16, IFIX/PYHIN1, and MNDA), and mice a total of 13 (Figure 1). Despite the significant diversity among PYHIN proteins in mammals, bats are the only group reported to lack the entire PYHIN gene locus; only a truncated version of AIM2 has been identified in one bat species, *Pteronotus parnellii* [31]. Among PYHIN proteins, AIM2 is unique as it diverges significantly from other PYHIN proteins and is the only one with clear orthology across numerous species. Phylogenetic analyses of the HIN and Pyrin domains suggest that human and mouse AIM2 are the only direct orthologs [7]. While there are no direct orthologs of human IFI16, IFIX, or MNDA in mice, mouse p204, p205, p207, and p211 cluster with human IFI16 and IFIX in phylogenetic analyses [7,18], suggesting that they may exert similar activities. Gene duplications are commonly observed in immune genes and may allow diversification of expression patterns, targeting of newly emerging pathogens, or evasion of viral antagonists [32,33,34]. Mice are infected with a large variety of viral families, and it is tempting to speculate that the multiplication of Pyhin genes was driven by the need to defend and control a wide range of pathogens.

PYHIN proteins are characterized by an N-terminal Pyrin domain (PYD) and typically one C-terminal hematopoietic interferon-inducible nuclear protein with a 200-amino-acid repeat domain (HIN200) [1,2,7]. The PYD belongs to the superfamily of death domains (DDs) and promotes PYD-PYD interactions that regulate many cellular processes, including inflammation, apoptosis, and cell cycling [35]. The HIN domain is only found in PYHIN proteins and mediates non-specific DNA binding by electrostatic interactions between side chains of positively charged HIN domain amino acid residues and negatively charged phosphate groups in the DNA backbone [8,36]. HIN domains have diversified from a common precursor [7] and are designated -A, -B, and -C based on the amino acidic sequence following a conserved MFHATVAT motif [36].

A total of 10 of the 13 mouse PYHINs contain both a Pyrin and an HIN domain (Figure 2). Similar to human IFI16, p204 contains two HIN domains. They most likely arose from independent events since tandem HIN domains are not found in other non-primate species [7]. Structurally, p204 resembles human IFI16, p209 is similar to IFIX, p211 and p212 look like human MNDA, and p210 is the ortholog of human AIM2 (Figure 2). While the PYD and HIN domains are usually preserved, mouse PYHIN protein p202 lacks a PYD, and p208 and p213 lack an HIN domain. Mouse PYHIN proteins show an enormous variability in their size, amino acid sequences, and structural organization. Especially, the linker region between the PYD and HIN domains varies enormously. While relatively little is known about the function of the linker region, it contains one or two nuclear localization signals (NLSs) in human IFI16, IFIX, and MNDA, and these were critical for their antiviral activity [23,37]. NLS prediction analyses suggest that murine p203, p206, p207, p208, p209, p212, and p214 also contain at least one nuclear localization motif in their linker region (Figure 2). Notably, it has been reported that the linker region is under positive selection pressure, suggesting a potential role in pathogen interaction and innate defense mechanisms [38]. Altogether, mouse PYHIN proteins show enormous structural variability, indicating functional diversification.

### 3.2. Overexpression of Most Mouse PYHIN Proteins Inhibits HIV-1

For functional analyses, we generated constructs co-expressing the blue fluorescent protein (BFP) and HA-tagged versions of all 13 murine PYHIN proteins via an internal ribosome entry site (IRES). For comparison, we also examined human IFI16, MNDA, PYHIN1/IFIX, and AIM2, which have been previously analyzed for their antiretroviral activity [23,37,39]. Western blot analysis confirmed that all murine PYHIN proteins were expressed at detectable levels in transfected HEK293T cells (Figure 3a). However, expression levels varied substantially, and in some cases, products of various sizes were observed, indicating potential post-translational modifications and proteolytic processing. To determine their antiviral activity, HEK293T cells were cotransfected with vectors expressing the various PYHIN proteins and the proviral HIV-1 CH058 transmitted-founder (TF) construct [27]. We selected the infectious molecular CH058 clone for analyses because primary TF viruses are highly relevant for HIV-1 infection in vivo, and CH058 is efficiently inhibited by human nuclear PYHIN proteins [23,37]. As expected, the three nuclear human PYHIN proteins reduced the production of infectious HIV-1 particles, while AIM2 had only modest inhibitory effects (Figure 3b). A total of 8 of the 13 mouse PYHIN proteins (p203, p204, p205, p208, p209, p210, p211, and p212) inhibited HIV-1 with similar efficiency as human IFI16, PYHIN1, and MNDA proteins (Figure 3b). In comparison, p202b lacking a Pyrin domain, the two largest mouse PYHIN proteins (p206 and p207), as well as the shortest (p213 lacking an HIN domain), had little if any inhibitory effect. Notably, mouse p210 reduced infectious HIV-1 production more efficiently than its human AIM2 ortholog.

Human PYHIN proteins have been reported to inhibit HIV-1 by affecting viral transcription [23,37,39]. To determine if mouse PYHIN proteins may act in a similar manner, we analyzed the effect of human and mouse PYHIN proteins on the levels of HIV-1 *env* transcripts. In agreement with the effect on infectious virus yield, real-time quantitative PCR showed that nuclear human PYHIN proteins and the majority of mouse PYHINs suppress the production of HIV-1 *env* RNA transcripts in transfected HEK293T cells about as efficiently as human IFI16 (Figure 3c). Human AIM2 and its mouse ortholog p210, as well as p202, p212, p213, and p214, displayed no significant inhibitory effects on *env* transcripts. On average, p202 even increased the levels of HIV-1 *env* transcripts by ~2.5-fold. Antiviral activity did not correlate significantly with detectable PYHIN expression levels, and p203, p208, p209, p210, p212, and p214 inhibited HIV-1 despite relatively low expression (Figure 3d). However, all four PYHYIN proteins that were efficiently expressed (p204, p205, p211, and MNDA) inhibited HIV-1, while 4 of the 13 factors that were expressed at lower levels (p202, p206, p207, and p213) lacked antiviral activity. In addition, we only observed a non-significant trend between the effects of the PYHIN proteins on infectious HIV-1 yields and the expression of *env* mRNAs (Figure 3e). In contrast, the correlation between infectious virus and p24 capsid antigen yields in the cell culture supernatants was highly significant (Figure 3f). Altogether, these results showed that mouse PYHIN proteins inhibit HIV-1 production and further suggested that functional properties, as well as expression levels, play a role in their antiviral activity.

Next, we examined the subcellular localization of mouse PYHIN proteins in transiently transfected HEK293T (Appendix A). Confocal immunofluorescence microscopy analysis was performed by staining the C-terminal HA-tag of the PYHIN proteins, while the nucleus was stained by DAPI. Most antiviral mouse PYHINs, such as p203, p204, p205, and p211, were mainly detectable in the nucleus or the nuclear membrane (p208, p209) of HEK293T cells (Appendix A). However, some exceptions were identified; e.g., p210 and p214 were detected in the cytoplasm but showed antiviral activity, while p202 and p207 were nuclear but poorly antivirally active. Finally, p212 was detectable in both the cytoplasm and the periphery of the nuclei. Altogether, there was no definitive correlation between localization and antiviral activity. However, the diverse localization further supports that mouse PYHIN proteins may act at different sites and by various mechanisms.

To obtain further insight into the mechanism underlying the inhibitory effect of mouse PYHIN proteins on infectious HIV-1 production, we examined their effect on the activity of an HIV-1 NL4-3 LTR firefly luciferase reporter construct [23]. Consistent with published data [23,37], the three nuclear human PYHIN proteins, IFI16, MNDA, and IFIX, reduced HIV-1 LTR-driven luciferase activity by ~70 to 80% (Figure 4a). Most mouse PYHIN proteins also inhibited HIV-1 LTR activity, with p204, p205, p207, p209, p210, p211, p212, and p214 being about as effective as the human PYHIN proteins. In agreement with their poor inhibitory effect on infectious HIV-1 yield (Figure 3b), p202, p206, and p213 displayed little if any effect on HIV-1LTR activity (Figure 4a). In contrast, human AIM2 and its mouse ortholog p210 inhibited LTR-driven luciferase expression by ~80% (Figure 4a). This came as a surprise and needs further investigation since at least human AIM2 is predominantly found in the cytoplasm and did not reduce the levels of HIV-1 CH058 *env* transcripts. Altogether, the impact of the 4 human and 13 mouse PYHIN proteins on HIV-1 NL4-3 LTR promoter activity correlated significantly with their effects on infectious HIV-1 CH058 yields (Figure 4b) and (less well) the expression levels of *env* transcripts (Figure 4c). However, correlations were far from perfect, suggesting that the inhibition of viral transcription plays an important but not the only role in the inhibitory effects of mouse and human PYHIN proteins.

### 3.3. Mouse PYHIN Proteins Inhibit Murine Leukemia Virus (MLV)

To determine the restriction of a murine retrovirus, we cotransfected HEK293T cells with PYHIN expression vectors and an *env*-defective proviral Moloney MLV construct containing an eGFP reporter gene. The four human PYHIN proteins and most mouse PYHIN proteins (except p202, p206, and p213) suppressed LTR-driven eGFP expression by the Moloney MLV proviral construct, albeit with differential efficacy (Figure 5a). To clarify whether mouse PYHINs interfere with the LTR-driven transcription of mouse retroviruses, we examined their ability to inhibit luciferase expression driven by a Friend MLV LTR construct (Appendix A). All four human PYHIN proteins and 10 of the 13 mouse PYHIN proteins significantly inhibited Friend MLV (FV) LTR activity by 50% to 85% (Figure 5b). Altogether, the inhibitory effects of the different human and murine PYHIN proteins on Moloney MLV-driven proviral eGFP expression correlated significantly with those on Friend MLV LTR-dependent luciferase expression (Figure 5c). Similarly, PYHIN proteins inhibiting transcription of mouse retroviruses also inhibited HIV-1 LTR activity (Figure 5d,e). Notably, the effect of the PYHIN proteins on eGFP expression by the proviral MLV construct also correlated significantly with their effect on infectious HIV-1 production (Figure 5f). Altogether, these significant but imperfect correlations suggest that human and mouse PYHIN proteins inhibit mouse and human retroviruses by overlapping mechanisms.

We have previously shown that nuclear PYHIN proteins target the transcription factor Sp1 to restrict HIV-1 transcription [23,37,39]. To further elucidate the antiretroviral activity of mouse PYHIN proteins, we examined whether Sp1 also affects the gene expression of MLV. Sp1 overexpression in transfected HEK293T cells strongly increased LTR-driven GFP expression by the *env*-defective M-MLV proviral construct (Figure 6a). In agreement with the presence of Sp1 interaction sites in the F-MLV LTR (Appendix A), Sp1 increased luciferase gene expression in a dose-dependent manner (Figure 6b). To further examine the role of Sp1 in the inhibitory effects, we examined an HIV-1 LTR luciferase construct containing mutated Sp1 binding sites [23]. Despite the lack of Sp1 sites, most human and murine PYHIN proteins significantly inhibited LTR-driven luciferase activity (Figure 6c). As discussed below, this came as a surprise because we have previously shown that a lack of Sp1 binding sites significantly reduces the inhibitory effect of IFI16 on basal and Tat-induced LTR-driven gene expression [23].

## 4. Discussion

In the present study, we show that most of the 13 different mouse PYHIN proteins share the ability of nuclear human PYHIN proteins to inhibit human and mouse retroviruses. The underlying mechanisms and structural determinants of the antiretroviral activity of mouse PYHIN proteins need further study and seem to be complex. In most cases, the inhibitory effects on infectious HIV-1 yield correlated with the suppression of HIV-1 and Friend MLV LTR activity. However, some exceptions were observed; e.g., p206 inhibited infectious virus production on average by ~50% but had little effect on LTR activity. Vice versa, p207 reduced LTR activity by ~70% but infectious virus yields only by ~20%. In addition, p210 and p212 significantly reduced infectious virus yield (Figure 3b), although they had no significant effects on the levels of viral *env* transcripts (Figure 3c). Altogether, most but not all of the inhibitory effects of mouse PYHIN proteins in transfected HEK293T cells seem to involve the suppression of retroviral transcription. Our result that most mouse PYHIN proteins share the antiretroviral activity of human nuclear PYHIN proteins agrees with the previous finding that mouse PYHIN proteins were dispensable for the type I interferon (IFN) response to transfected DNA ligands, as well as infection with DNA and RNA viruses [22], but associated with reduced control of Friend MLV replication early after infection [23]. Our findings add to the evidence that PYHIN proteins are important effectors of the innate immune defense.

We have previously shown that the ability of human nuclear PYHIN proteins to inhibit HIV-1 involves interactions of their PYHIN domains with the transcription factor Sp1 and nuclear localization [23,37,39]. It has been reported that all murine PYHIN proteins (except IFI210/AIM2, which is cytosolic) localize to the nucleus when expressed alone [18]. However, many of them relocalize to the cytoplasm in the presence of STING and the ASC (apoptotic speck-like protein containing a caspase recruitment domain) protein [18]. In HEK293T cells, we detected most mouse PYHIN proteins in the nucleus, although their distribution varied: diffuse distribution throughout (p202, p203, p204, p205, and p206), periphery or nuclear membrane (p208, p209), and punctate pattern (p207, p213). P212 and p214 were detected in both the cytoplasm and the nucleus, while p210 was only found in the cytoplasm (Appendix A). Overexpression may cause artifacts, and further studies of endogenously expressed mouse proteins are required to define potential correlations between subcellular localization and antiretroviral activity. Sp1 increased the transcription of Moloney and Friend MLV (Figure 6) and preliminary results from co-immunoprecipitation and proximity ligation assays suggest that some mouse PYHIN proteins may interact with Sp1. Thus, it came as a surprise that a lack of the Sp1 binding sites had little impact on the susceptibility of the HIV-1 LTR to inhibition by PYHIN proteins. Our previous studies using a large variety of HIV-1 constructs and experimental systems clearly demonstrated that sequestration of Sp1 plays a key role in the antiretroviral activity of nuclear human PYHIN proteins [23,37,39]. The exact reasons why this could not be recapitulated in the present transient transfection assays using LTR luciferase constructs needs further investigation. In addition, further analyses in more relevant cell types are required to elucidate the mechanisms underlying the antiretroviral activities of mouse PYHIN proteins and dissect them from their effects on STING-dependent IFN production and activation of the ASC inflammasome. Notably, the PYHIN domains of mouse p204, p205, p207, and p211 cluster with the Pyrin domains of human IFIX and IFI16 in phylogenetic analyses, and these mouse PYHIN proteins are strongly expressed in the skin [7,18]. They all significantly inhibited retroviral LTR activity and (except p207) infectious HIV-1 production. Defense against invading viral pathogens is thus a plausible function for this group of human and mouse PYHIN proteins.

Only 3 of the 13 mouse PYHIN proteins (p202, p206, and p213) did not display any significant inhibitory effects on gene expression by a proviral Moloney MLV construct and the Friend MLV LTR (Figure 5a,b). Notably, p202 lacks a Pyrin domain [7,18]. In agreement with the distinct role of p202 in innate immunity, it has been reported that p202 is an inhibitor of DNA-induced caspase activation [40]. p202 lacks nuclear localization signals and resides in the cytoplasm of untreated cells [40,41], although one study also detected it in the nucleus [18]. Similarly, p206 has been reported to have a cytoplasmic location [42]. Our results that p202 and p206 do not inhibit retroviral transcription agree with the previous findings that these are usually not found in the nucleus. Finally, p213 is a very short protein of just 135 amino acids that lacks an HIN domain. Interestingly, it has been reported that p213 (PYR-A) is the strongest activator of STING of all murine PYHIN proteins [18]. Notably, p208 also lacks an HIN domain significantly inhibited by LTR-dependent gene expression and production of HIV-1. This agrees with our previous finding that the HIN domains of IFI16 are dispensable for antiretroviral activity [23,37].

AIM2 and p210 are the only direct orthologs of PYHIN proteins between humans and mice [7,18]. It has been shown that mouse AIM2 is required for effective control of *Francisella tularensis* and MCMV infection of mice [43,44,45,46]. In agreement with the key role of mouse AIM2/p210 in inflammasome activation, AIM2-deficient mice showed reduced IL-1β production and pyroptosis in response to several viruses and intracellular bacteria [42,43,44,45]. In contrast, all mouse PYHIN proteins were dispensable for IFN activation in response to virus infection and DNA ligands [22], which agrees with minimal STING-dependent activity and an intact interferon-stimulatory DNA (ISD) pathway in AIM2-deficient murine cells [41]. Altogether, these results show that the AIM2 ortholog in mice activates the inflammasome but does not stimulate the IFN response. Human AIM2 and mouse p210 had no significant effects on the levels of HIV-1 *env* transcripts (Figure 3c) but suppressed luciferase gene expression from HIV-1 and Friend MLV LTRs. This was unexpected, and further studies are required to clarify if the latter effects are due to inflammasome activation or other mechanisms.

One important limitation of the present study is that the results were obtained by transient overexpression in HEK293T cells. Notably, however, it has been previously reported that all 13 murine proteins are IFN-inducible and also activated by nucleic acid ligands [18]. In transient transfection assays, most of them were expressed at similar levels as IFI16 (Figure 3a) and achieved similar efficiencies in inhibiting HIV-1 and MLV. For human IFI16, it has been shown that knock-down or knock-out in primary macrophages and T cells substantially increases HIV-1 replication [23,37]. While additional work is necessary to obtain further definitive proof, these results suggest that mouse PYHIN proteins display significant antiviral activity independently of immune sensing and inflammasome activation.

It is conceivable that the mouse PYHIN proteins multiplied to inhibit pathogens at different locations and by different mechanisms. Our results contribute to the evidence of functional redundancy and the remarkable diversification of mouse PYHIN proteins. For example, it has been reported that p202, p203, p207, and p213 all promote STING-dependent IFN activation, while p210, p211, and p212 act as robust activators of the ASC inflammasome [18]. Most of the remaining mouse PYHIN proteins (i.e., p204, p205, p208, p209, and p214) significantly inhibited retroviral transcription and HIV-1 replication. In comparison, p214 was less active than p209, which has a highly similar structure. However, p214 was only expressed at low levels (Figure 3a). While further work is required, these results clearly suggest that mouse PYHIN proteins have overlapping but distinct functions in inflammasome activation, IFN induction, and more direct suppression of viral pathogens.

## Figures and Tables

**Figure 1 viruses-16-00493-f001:**
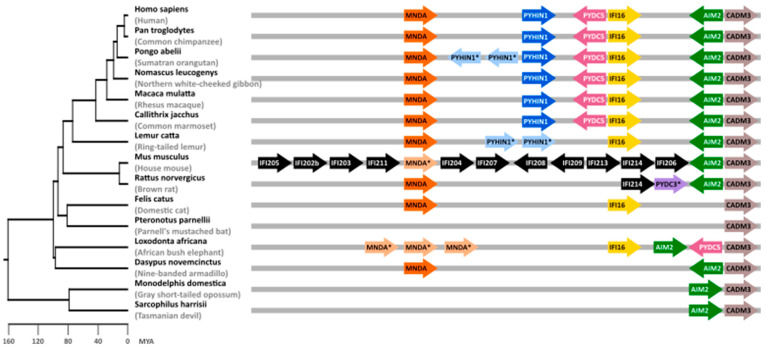
Mammalian PYHIN gene locus. Syntenic blocks containing PYHIN paralogs from selected mammalian species are shown on the right. Arrows indicate the direction of the open reading frames (ORFs). Genes annotated as orthologs are shown in identical colors. Genes annotated as PYHIN1-, MNDA-, or PYDC3-like genes are marked with an asterisk “*”. Note that identical names do not necessarily indicate true orthologs. In mice, most genes are shown in black as it is not possible to clearly identify orthologs in other species. CADM3 has been included as a reference gene flanking the PYHIN gene locus. Gene lengths, distances, and overlaps are not drawn to scale, and some overlapping and additional ORFs were omitted for clarity. A time-calibrated phylogenetic tree of mammalian evolution generated with TimeTree [26] is shown on the left. MYA—million years ago.

**Figure 2 viruses-16-00493-f002:**
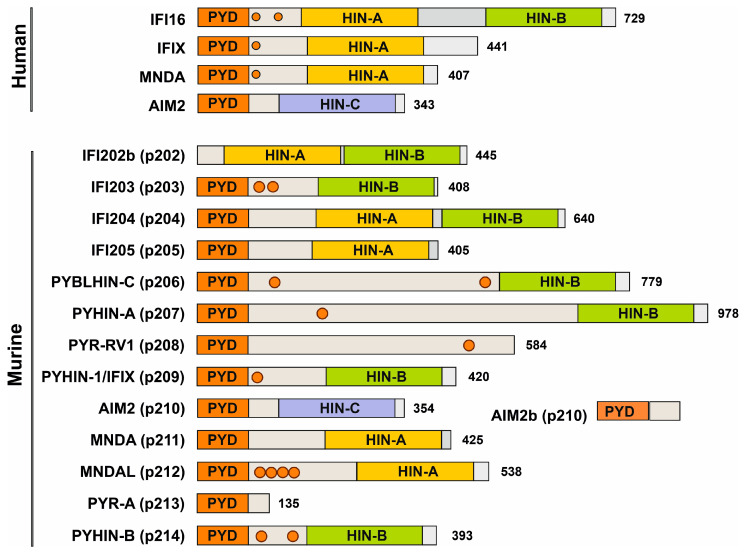
Domain organization of human and murine PYHIN proteins. Most members of the PYHIN protein family are characterized by a Pyrin domain (orange) at their N-terminus and a C-terminal HIN domain. Structural variations within the HIN domain delineate it into three subtypes: HIN-A, HIN-B, and HIN-C (shown in yellow, green, and purple, respectively) and a linker region (gray) with potential nuclear localization signals (NLSs) indicated with circles. Modified from Cridland and colleagues [7].

**Figure 3 viruses-16-00493-f003:**
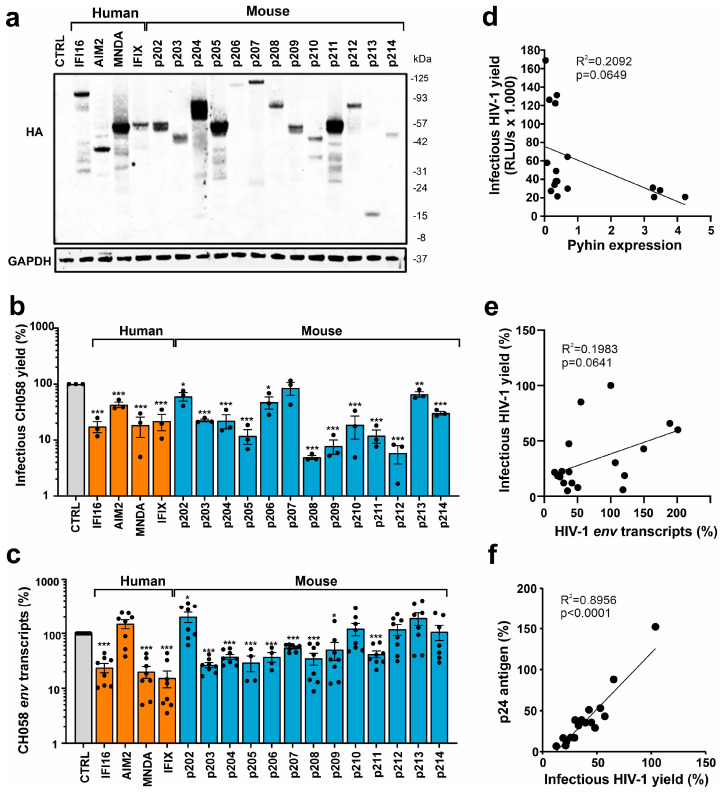
Expression of PYHIN proteins and inhibition of HIV-1. (**a**) Expression of human and mouse PYHIN proteins. HEK293T cells were transfected with IRES-BFP constructs expressing the indicated HA-tagged PYHIN proteins. GAPDH was used as loading control. (**b**) Infectious virus yields in cell supernatants of HEK293T cells cotransfected with vectors expressing the indicated PYHIN proteins and the proviral HIV-1 CH058 construct were determined by TZM-bl cell infection assay at two days post-transfection. Data show mean percentages (±SEM; *n* = 3). (**c**) Levels of viral env transcripts in HEK293T cells cotransfected with expression constructs for the indicated PYHIN proteins and the proviral HIV-1 CH058 construct determined by qRT-PCR. Data show mean percentages (±SEM) from 3 experiments, each performed in technical duplicates relative to those detected in the presence of the empty control plasmid (100%). * *p* < 0.05, ** *p* < 0.01, *** *p* < 0.001. (**d**–**f**) Correlations between the effects of overexpression of PYHIN proteins on infectious HIV-1 yield and (**d**) the levels of PYHIN proteins detected by Western blot, (**e**) *env* RNA transcripts, and (**f**) p24 antigen levels. Values in panels d and e were derived from the results shown in panels (**a**–**c**).

**Figure 4 viruses-16-00493-f004:**
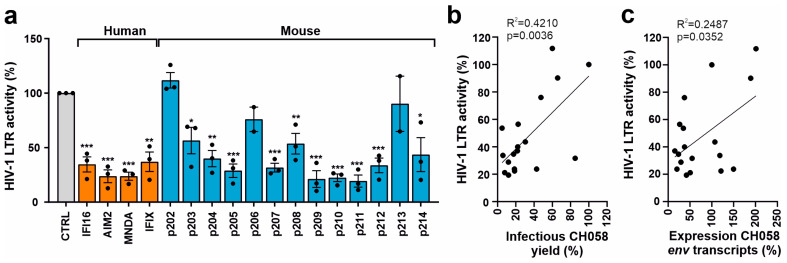
Effect of human and mouse PYHIN proteins on basal HIV-1 LTR activity. (**a**) HEK293T cells were cotransfected with a construct containing the firefly luciferase reporter gene under the control of the HIV-1 NL4-3 LTR. Values show mean percentages (±SEM) relative to those detected in the absence of PYHIN protein (100%) and were derived from three experiments. * *p* < 0.05, ** *p* < 0.01, *** *p* < 0.001. (**b**,**c**) Correlation between the effects of PYHIN protein overexpression on HIV-1 LTR activity and (**b**) infectious HIV-1 yield and (**c**) env RNA levels.

**Figure 5 viruses-16-00493-f005:**
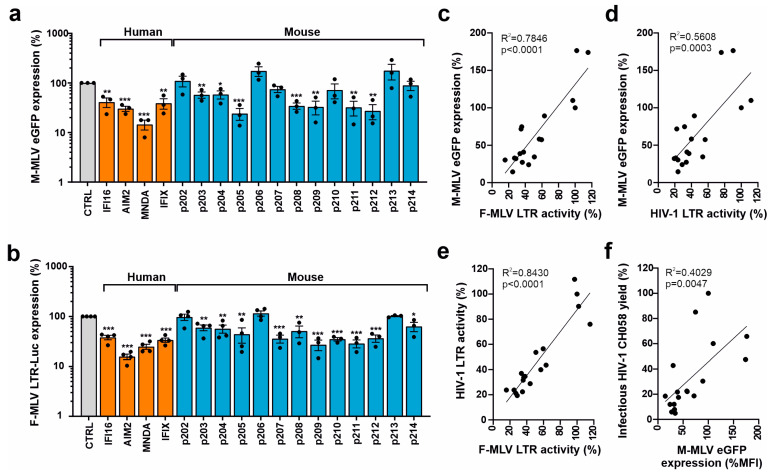
Effect of human and mouse PYHIN proteins on MLV transcription and LTR activity. (**a**) HEK293T cells were cotransfected with either a control vector or plasmids expressing HA-tagged mouse and human PYHIN proteins in a pCG IRES BFP vector together with an env-defective proviral Moloney MLV (M-MLV) eGFP construct. Shown is the eGFP mean fluorescence intensity (MFI) in BFP/GFP double-positive cells. Shown are mean values (±SEM) from three independent experiments in the presence of PYHINs relative to the empty vector control (100%). * *p* < 0.05, ** *p* < 0.01, *** *p* < 0.001. (**b**) HEK293T cells were cotransfected with a Friend MLV (F-MLV) LTR luciferase reporter construct and vectors expressing the indicated PYHIN proteins. Values show mean percentages (±SEM) relative to those detected in the absence of PYHIN protein (100%) and were derived from three or four experiments. (**c**–**f**) Correlation between the effects of overexpression of mouse and human PYHIN proteins on M-MLV-driven eGFP expression, F-MLV and HIV-1 LTR activity, and infectious HIV-1 yield as indicated.

**Figure 6 viruses-16-00493-f006:**
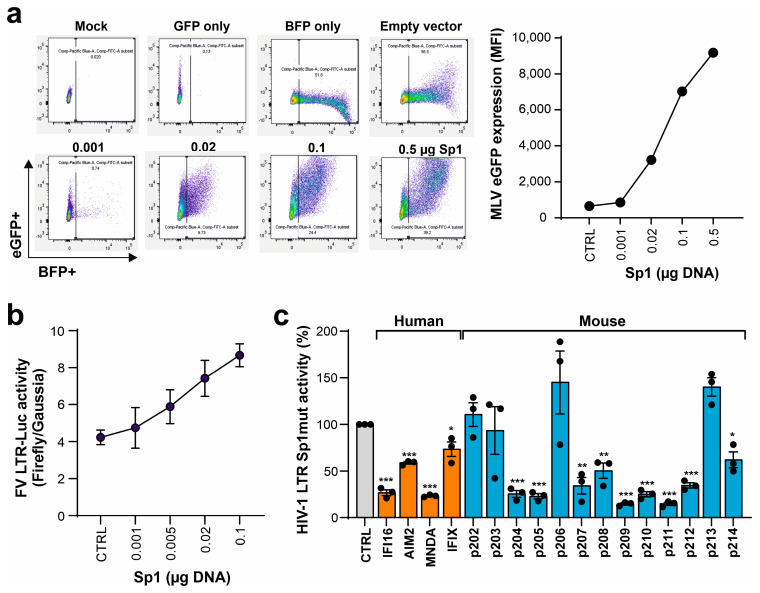
Sp1 enhances MLV gene expression but is not essential for inhibition by mouse PYHIN proteins. (**a**) HEK293T cells were cotransfected with increasing amounts of expression constructs for Sp1 IRES BFP (0.001, 0.02, 0.1, and 0.5 µg) and MLV eGFP construct. BFP and eGFP expression was analyzed by flow cytometry. The left panel shows primary FACS data, and the right panel shows the mean fluorescence intensities (MFIs) of eGFP in BFP expressing population. (**b**) HEK293T cells were cotransfected with a firefly luciferase reporter construct under the control of FV LTR and a vector expressing an increasing amount of Sp1 (0.001, 0.005, 0.02, and 0.1 µg) or an empty control. In addition, a Gaussia luciferase reporter construct was transfected as an internal control. Shown are firefly relative to Gaussia luciferase values measured in the presence of Sp1 overexpression; *n* = 3 ± SEM. (**c**) HEK293T cells were cotransfected with an NL4-3 LTR firefly luciferase reporter construct containing mutated Sp1 binding sites and expression constructs for the indicated PYHIN proteins. Values show mean percentages (±SEM) of firefly luciferase expression relative to those detected in the absence of PYHIN protein (100%) and were derived from three experiments. * *p* < 0.05, ** *p* < 0.01, *** *p* < 0.001.

## Data Availability

All primary data will be available from the corresponding author upon request.

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
