# Peer review of "A Variety of Mouse PYHIN Proteins Restrict Murine and Human Retroviruses"

_viruses, 2024, doi:10.3390/v16040493_

Round 1

Reviewer 1 Report

Comments and Suggestions for Authors

The study from Erdemci-Evin and colleagues examines the effects of 13 murine PYHIN proteins against HIV-1 and MLV. The study uses co-expression of the different PYHIN proteins along with HIV-1 or MLV proviral constructs to examine infectivity of released particles and LTR activity and concludes on the large variability of the effects of these proteins and virus inhibition.

The experiments are well conducted and the question is of interest in light of the large expansion of the PYHIN family in mice. The study would merit from a deeper analysis of the results in terms of:

1)    Correlation between the antiviral activity of the different orthologues and their intracellular expression levels

2)    Correlation between the antiviral activity of the different orthologues and their intracellular distribution by confocal microscopy, which should be feasible in light of the fact that they are fused with BFP.

3)    Correlation between LTR activities with respect to the activity on non-viral, but SP1-containing cellular promoters (as the authors have already implicated SP1 in the antiviral activity of IFI16: doi:10.1016/j.chom.2019.05.002).

The latter could reveal possible discriminant differences that would be important to ascertain and would perhaps better increase our understanding of the different roles that these murine PYHIN may play

Reviewer 2 Report

Comments and Suggestions for Authors

“A variety of mouse PYHIN proteins restrict murine and human retroviruses” by Erdemci-Evin et.al

In this paper, the authors examined the antiretroviral activity of 13 mouse PYHIN proteins in inhibiting HIV-1 and murine leukemia viruses (MLVs). These proteins are known to play a complex role in innate immunity, with most of them acting as activators of STING-dependent IFNs and/or ASC-dependent inflammasome pathways against bacterial and viral pathogens. It was previously shown by this group that human nuclear PYHIN IFI16 targets the transcription factor Sp1 to suppress HIV-1 transcription, demonstrating a more direct interaction with the virus. Here they show that transient overexpression of mouse PYHIN proteins inhibit, to varying degrees, HIV-1 production and expression of env transcripts. Furthermore, PYHIN proteins exert an inhibitory effect on Moloney MLV-driven proviral eGFP expression. The authors also report inhibition of HIV-1 and Friend MLV LTR activity. Although many of the questions raised by these observations are far from being answered, some data are intriguing and novel. However, there are some issues to consider before publishing.

Major comments

1.  Materials and Methods - Luciferase Assay: can the authors specify which firefly luciferase reporter constructs are used?

2.  Experiments on the inhibition of LTR activity should be better described in the Materials and Methods or in the figure legends. In particular, for HIV/MLV LTR firefly luciferase reporter constructs, are they expressed in HEK293T at basal levels or do they require a trans activator (e.g. TAT, virus)? By what mechanism do the authors hypothesize that mouse PYHIN proteins inhibit LTR activity?

3. Figure 3b shows that p206 and p207 have little if any inhibitory effect on HIV-1 virus production (lines 247-49), so the statement in the discussion that “However, some exceptions were observed, e.g. p206 inhibited infectious virus production more efficiently than LTR activity, while the opposite was observed for p207” is not clear.

4. Lines 466-67: “Our preliminary data show that efficient transcription of Moloney and Friend MLVs is dependent on Sp1 and suggest that some mouse PYHIN proteins may interact with Sp1.” Are these “unpublished data”? It is unclear whether the authors allude to the LTR inhibition results reported in this manuscript. If so, this statement should be softened. Although the authors have previously demonstrated that IFI16 targets the transcription factor Sp1, they currently can only speculate that mouse PYHIN proteins act by the same mechanism.  

5. Do the authors have any additional information about p24 or RT activity of viruses obtained in presence of PYHIN proteins?

6. Some parts of the manuscript need to be improved in terms of writing.

Minor comments

1. Line 146: “40” should be “Forty”

2. Line 186: spelling of “typically” is wrong.

3. Line 230: 3.3 should be 3.2

4. Line 269: reference 24? It is not relevant.

5. Line 278: Figure 3c should be Figure 3d

6. Line 308: 3.4 should be 3.3

7. Line 379: Figure 4 should be Figure 5a,b

8. Line 386: the sentence “p213 is a very short protein of just 135 amino acids that lacks a pyrin domain.” Should be “…that lacks a HIN domain.”

9. Reference 29 is missing from the text.

Comments on the Quality of English Language

Minor editing of the English language is required.

Round 2

Reviewer 1 Report

Comments and Suggestions for Authors

The authors have addressed the comments of this referee

Reviewer 2 Report

Comments and Suggestions for Authors

The authors have adequately addressed my comments in the revised version of the manuscript. Therefore, I have no further comments.